# Post-insemination selection dominates pre-insemination selection in driving rapid evolution of male competitive ability

**Katja R. Kasimatis**[¤]*, **Megan J. Moerdyk-Schauwecker**, **Ruben Lancaster**, **Alexander Smith**, **John H. Willis**, **Patrick C. Phillips***

Institute of Ecology and Evolution, University of Oregon, Eugene, Oregon, United States of America

¤ Current address: Department of Ecology and Evolutionary Biology, University of Toronto, Toronto, Ontario, Canada
* k.kasimatis@utoronto.ca (KRK); pphil@uoregon.edu (PCP)

**Data Availability Statement:** The oligonucleotides and synthetic constructs used in this study are available as S1–S3 Tables. Sequence data are

## Abstract

Sexual reproduction is a complex process that contributes to differences between the sexes and divergence between species. From a male's perspective, sexual selection can optimize reproductive success by acting on the variance in mating success (pre-insemination selection) as well as the variance in fertilization success (post-insemination selection). The balance between pre- and post-insemination selection has not yet been investigated using a strong hypothesis-testing framework that directly quantifies the effects of post-insemination selection on the evolution of reproductive success. Here we use experimental evolution of a uniquely engineered genetic system that allows sperm production to be turned off and on in obligate male-female populations of *Caenorhabditis elegans*. We show that enhanced post-insemination competition increases the efficacy of selection and surpasses pre-insemination sexual selection in driving a polygenic response in male reproductive success. We find that after 10 selective events occurring over 30 generations post-insemination selection increased male reproductive success by an average of 5- to 7-fold. Contrary to expectation, enhanced pre-insemination competition hindered selection and slowed the rate of evolution. Furthermore, we found that post-insemination selection resulted in a strong polygenic response at the whole-genome level. Our results demonstrate that post-insemination sexual selection plays a critical role in the rapid optimization of male reproductive fitness. Therefore, explicit consideration should be given to post-insemination dynamics when considering the population effects of sexual selection.

## Author summary

Some of the most dramatic and diverse phenotypes observed in nature––such as head-butting in wild sheep and the elaborate tails of peacocks––are sexually dimorphic. These remarkable phenotypes are a result of sexual selection optimizing reproductive success in females and males independently. For males, total reproductive success is comprised of winning a mating event and then translating that mating event into a fertilization event.

available at NCBI SRA under accession number PRJNA795674. Model summary statistics for the genomic analyses (S1–S6 Data) and the fertility data (S7 Data) are at figshare DOI:10.6084/m9.figshare.14902599. All R scripts are available via the GitHub repository https://github.com/katjakasimatis/postinsemination_expevol. Availability of materials: Worm strains PX624, PX631, and PX658 are available from the Caenorhabditis Genetics Center (https://cgc.umn.edu). All other strains are available from the Phillips Lab upon request (email pphil@uoregon.edu; https://pages.uoregon.edu/pphil/index.html).

**Funding:** This work was funded by National Institutes of Health grant R35GM131838 to PCP (https://grants.nih.gov/grants/guide/pa-files/PAR-17-094.html). KRK is supported by a Natural Sciences and Engineering Research Council of Canada Banting Postdoctoral Fellowship (https://banting.fellowships-bourses.gc.ca/en/2019-2020-eng.html). The funders had no role in study design, data collection and analysis, decision to publish, or preparation of the manuscript.

**Competing interests:** The authors have declared that no competing interests exist.

Therefore, to understand not only how male reproductive success is comprised, but also how it evolves, we must examine the interaction between pre- and post-insemination sexual selection. We combine environmentally-inducible control of sperm production within a highly reproducible factorial experimental evolution design to directly quantify the contribution of post-insemination selection to male reproductive evolution. We demonstrate that enhanced sperm competition increases the efficacy of selection and enhances the rate of male evolution. Alternatively, we show that enhanced pre-insemination competition slows the evolutionary rate. Using whole-genome approaches, we identify over 60 genes that contribute to male fertilization success. Brought together, our new approaches and results demonstrate that the unseen world of molecular interactions occurring during post-insemination are as fundamentally important as pre-mating factors.

## Introduction

Sexual selection drives the evolution of some of the most remarkable phenotypes observed in nature. It is tempting to focus on these flashy phenotypes involved in pre-fertilization reproductive dynamics, such as male-male competition and female choice [1]. However, in animals with internal fertilization, reproduction is more complex and requires a series of interactions within and between the sexes to produce a viable offspring. From a male's perspective, total reproductive success can be partitioned into winning a mating event and then winning a fertilization event. Therefore, sexual selection has the potential to act on both the variance in mating success and the variance in fertilization success (also referred to as gametic selection [2,3]). Whether selection during these reproductive phases interacts in an additive, antagonistic, or synergistic manner to optimize total male reproductive success is understudied. Understanding this balance is critical for understanding how sexual selection shapes the evolution of reproductive success over time. Such processes are critical for relating the role of sexual selection to population adaptation [4,5] and divergence [6].

Experimental separation of sexual selection before and after mating within an adaptive framework has proved extremely challenging. A meta-analysis of studies across 21 different taxa found no significant relationship between secondary sexual traits and ejaculate quality [7]. Alternatively, previous studies have taken the approach of Arnold and Wade [8] to partition the variance in total reproductive success into the variance in mating success and the variance in fertilization success, reviewed in [9]. These studies have inferred mixed results as to the opportunity for sexual selection. Several studies suggest that the variance in mating success comprises greater than 95% of the total variance in reproductive success [10,11], while others indicate a greater contribution of the post-insemination phase [12–16]. Additionally, evolutionary analyses of seminal fluid proteins show a high opportunity for post-insemination selection [17,18] and gamete proteins in broadcast spawning organisms support strong gametic selection [19]. While informative, the opportunity for sexual selection does not necessarily translate into realized selection, which contributes to the lack of consistent patterns between studies. Moreover, this framework is an indirect approach for partitioning reproductive success and thus lacks the ability to connect the action of selection to the underlying genomic response to understand how reproductive success is evolving.

*Caenorhabditis elegans* is an ideal system for disentangling mating interactions. First, the mating system in *C. elegans* can be manipulated to prevent hermaphrodite self-sperm production and create functional females that rely on male-female mating. Males in these functional female-male populations have low reproductive success relative to males from obligate

outcrossing *Caenorhabditis* species [20], which creates a high opportunity for the evolution of reproductive success. Second, we have developed an external, non-toxic sterility system for *C. elegans* [21] that capitalizes on the auxin-inducible degron system to degrade the critical spermatogenesis gene *spe-44* and effectively turn off sperm production. The induction of sterility allows for sperm competitive dynamics to be isolated from male-male competitive dynamics for thousands of worms at a time. Finally, *C. elegans* is amenable to the evolve and re-sequence experimental approach [22,23], which allows us to not only quantify the impact of sexual selection on reproductive success, but also identify the underlying genetic structure of the traits involved.

Here we capitalize on transgenics to isolate the contributions of pre-insemination mating competition versus post-insemination sperm competition to the evolution of reproductive fitness of a newly derived *C. elegans* male population. We first create an obligate outcrossing *C. elegans* population composed of functional females and males with inducible sterility. We then performed 30 generations of replicated experimental evolution using a factorial design that partitions sexual selection due to within-strain and between-strain competitive dynamics occurring during pre-insemination and post-insemination. This experiment explicitly tests if pre-insemination sexual selection and post-insemination sexual selection contribute to reproductive success in an additive, synergistic, or antagonistic manner. If pre- and post-insemination selection are additive or synergistic, then we expect to see the greatest increase in total reproductive success when competition is enhanced through the addition of external male competitors during both reproductive stages. Alternatively, if these phases are antagonistic such that competition is beneficial during one stage but detrimental during the other, then we expect to see a reduction in total reproductive success when competition is enhanced during both reproductive stages. We can infer the source of antagonistic competition by comparing-and-contrasting the effects of enhanced and reduced post-insemination competition.

## Results

### Factorial framework to isolate selection on mating and fertilization success

We designed an experimental evolution framework that controls pre- and post-insemination competitive interactions using three distinct and powerful genetic manipulations: a mutation in the sex determination pathway (*fog-2*) to disrupt self-sperm production in hermaphrodites and maintain obligate male-female mating [20], targeted degradation of a key spermatogenesis protein (*spe-44*) to control male mating duration [21], and an inducible lethal marker (*peel-1*) to eliminate offspring from competitor males [24]. We previously showed that large populations of males produced sufficient sperm during initial, larval spermatogenesis to inseminate females prior to becoming fully sterile after 24 hours of exposure to auxin [21] (S1 Fig). Since females can store sperm and continuously lay eggs for at least three days [25], the sterility induction tool allowed us to control the length of male-male competition and then isolate it from the period of sperm competition which contributed to offspring production, as follows. To generate a selective event, male sterility was induced after an initial mating period (Figs 1 and S1). Increased sperm competition was then generated by adding competitor males from a different strain. Since temporal order of sperm transfer does not impact sperm usage in *C. elegans* [26], sperm from experimental males and sperm from competitor males theoretically had equal access to fertilize oocytes. After a 24 hour competitive phase, eggs were collected, hatched, and then heat-shocked to induce lethality of the competitor male cross-progeny, leaving only those progeny from the experimental males to start the next generation. Thus, this design isolates sperm competitive success from male mating success and selects for sperm

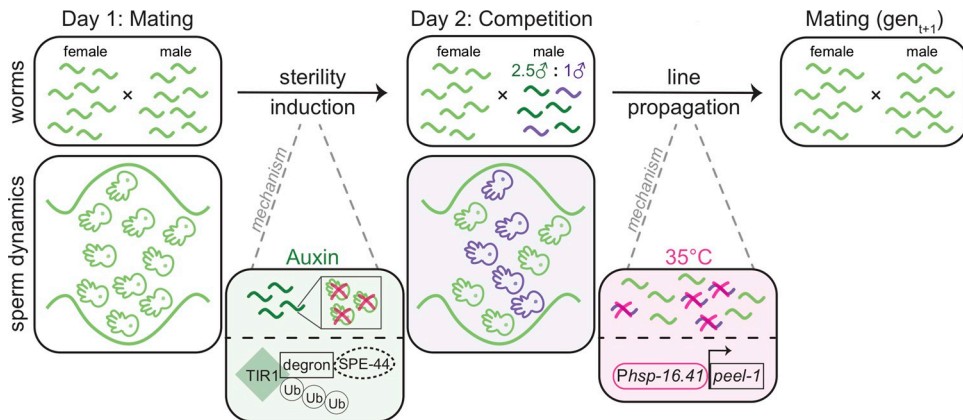

**Fig 1. Day-by-day depiction of the experimental evolution design shown at the population level and at the sperm level.** On day 1 sterility is induced by transferring worms to auxin-containing media. Auxin activates TIR1 to target the degron tag on SPE-44. The depletion of SPE-44 stops the production of sperm thereby inducing sterility. Females start laying eggs on day 1, using sperm from experimental males. On day 2, competitor males are added to the population at a ratio of 1 competitor male to 2.5 experimental males. Progeny are collected on day 3 from eggs laid on day 2 and then heat-shocked on day 4 to induce ectopic expression of the toxic protein PEEL-1. This expression kills competitor cross-progeny, leaving only the progeny from sperm transferred during the day 1 mating phase. Each selective event is followed by a recovery generation.

defensive capability and longevity. Additionally, as a byproduct of the inducible lethality marker, the design also selects for heat stress resistance.

The induction of sterility and addition of competitor males generated a factorial experimental design resulting in four experimental evolution regimes (Figs 2A and S1). When both sterility was introduced and competitors subsequently added (between-strain post-insemination only competition, BS-PO), there was increased sexual selection on post-insemination fertilization dynamics. Alternatively, when only sterility was induced, and competitor males not added (within-strain post-insemination only competition, WS-PO), experimental males experienced reduced sperm competition and potentially decreased post-insemination sexual selection. To represent the full degree of sexual selection acting on pre- and post-insemination competition (between-strain pre- and post-insemination competition, BS-P&P), sterility was not induced, but competitor males were added. Finally, no direct sexual selection was applied when neither sterility was induced nor competitors added (within-strain pre- and post-insemination competition, WS-P&P). The WS-P&P regime represents the base level of sexual selection experienced by recently derived *C. elegans* males. Due to the induction of sterility by day 2, both the WS-PO and BS-PO regimes experienced reduce pre-insemination selection relative to the WS-P&P and BS-P&P regimes. Therefore, by comparing the regimes we can partition the relative contributions of enhanced competition–and the opportunity for stronger selection–occurring during pre-versus post-insemination to changes in male reproductive success.

## Opportunity for selection is high in the ancestral population

We used multiple rounds of low-dose ethyl methanesulfonate (EMS) mutagenesis to generate genetic variation in the ancestral population of all four regimes (S2 Fig). Based on the mutation rate of EMS per generation [27], at least 937,500 non-exclusive mutations were expected to be segregating in the post-mutagenesis population prior to lab adaptation. We observed 321,929 SNPs segregating in the ancestral population, suggesting strong purifying selection during the pre-experimental evolution lab adaptation period (S2 Fig). The ancestral population had a genome-wide mean nucleotide diversity of $\pi = 0.06$ and the minor allele frequency ranged

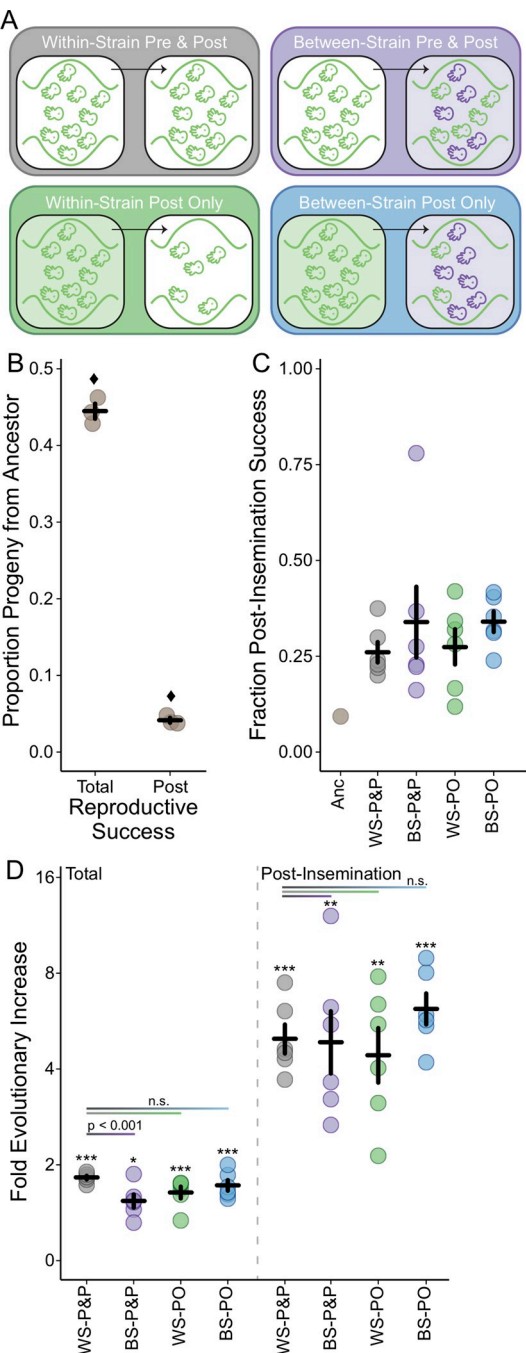

**Fig 2. The competitive reproductive success of males before and after experimental evolution under four sexual selection regimes. A)** Partitioning the sterility and competition treatments leads to four experimental evolution regimes: within-strain pre- and post-insemination competition (WS-P&P, gray), within-strain post-insemination only competition (WS-PO, green), between-strain pre- and post-insemination competition (BS-P&P, purple), and between-strain post-insemination only competition (BS-PO, blue). **B)** Ancestral males have poorer reproductive success than competitor males under both pre- and post-insemination competitive conditions (total) and under only post-insemination competitive conditions. Each point represents an independent assay with the mean and standard error across assays given. Diamonds denote a significant deviation from the null hypothesis of equal competitive ability between ancestral and competitive males for each condition (total: $\chi^2 = 6.87$, d.f. = 1, p < 0.01, 95% C.I. of ancestral competitive success = 40.4–48.6%; post-insemination: $\chi^2 = 863$, d.f. = 1, p < 0.0001, 95% C.I. of ancestral sperm competitive success = 3.0–5.5%). **C)** The fraction of total reproductive success attributable to post-insemination success in the ancestral population (Anc) and the evolved populations (G31: WS-P&P, BS-P&P, WS-PO, BS-PO). Each point represents a mean of three independent assays for the ancestor and each evolved replicate with the mean and

standard error across evolved replicates shown. **D)** The fold change in the total reproductive success and the post-insemination reproductive success of males in the evolved regimes relative to the ancestor (plotted on a $\log_2$ scale). Males in all regimes significantly increased in both measures of reproductive success (*p < 0.05, **p < 0.01, ***p < 0.001). Post hoc tests for a difference between the WS-P&P and the BS-P&P, WS-PO, and BS-PO regimes are indicated by the horizontal lines. The only significant difference appears between the total reproductive success of the WS-P&P and BS-P&P regimes, in which pre-insemination competition reduces the evolutionary response. Each point represents a mean of three independent assays for each evolved replicate with the mean and standard error across replicates shown.

from 0.004 to 0.5 (S3A–S3D Fig and S2 Data). These diversity estimates are higher than those commonly observed in *C. elegans* and are more comparable to the obligate outcrossing species *C. remanei* [28]. The distribution of variants was uneven across chromosome domains, following the characteristic pattern of higher diversity on the chromosome arms when compared to the chromosome center (Kolmogorov-Smirnov test: D = 0.072, p < 0.001) [29–31] (S3C and S3D Fig and S2 Data). SNP density strongly reflected this chromosome arm-center pattern: the mean SNP density on chromosome arms was $\theta_w = 0.0005$ and in chromosome centers was $\theta_w = 0.0003$ (S3E and S3F Fig). Despite the X chromosome having a slightly higher recombination rate in the small chromosome center domain [29] coupled with a greater opportunity for purifying selection in males, the X did not have the lowest SNP density (mean $\theta_w = 0.00033$) as expected. Instead, chromosome I had a significantly lower mean SNP density (mean $\theta_w = 0.00028$; t = -29, p < 0.001) than the other chromosomes. Together these summary statistics indicate that the ancestral population had more segregating genetic variants than is commonly observed in *C. elegans*, though much of this diversity is not in the gene dense chromosome centers.

We quantified ancestral reproductive success under highly competitive conditions occurring during both pre- and post-insemination (i.e., total reproductive success) and during post-insemination alone using a novel male competitor (Fig 2B). Total reproductive success was slightly, though significantly, poorer than the null expectation of equal competitive ability between ancestral male and competitor male backgrounds (proportions test: $\chi^2 = 6.87$, d.f. = 1, p < 0.01, 95% C.I. of ancestral competitive success = 40.4–48.6%). Ancestral male sperm competitive ability was especially poor with an average of 4.1% of progeny coming from ancestral males relative to the competitor (proportions test: $\chi^2 = 863$, d.f. = 1, p < 0.0001, 95% C.I. of ancestral sperm competitive success = 3.0–5.5%). Therefore, in the ancestral population post-insemination success only contributed 9.2% to the overall reproductive success of males (Fig 2C). The poor reproductive success of ancestral males under competitive conditions indicates the opportunity for selection–particularly gametic selection–to improve male competitive ability was high.

## Post-insemination selection drove evolutionary change in males

We quantified total reproductive success for each replicate population after 10 selective events occurring over 30 generations of evolution under the same highly competitive conditions used to assay the ancestral males. The contribution of post-insemination increased across all evolved (G31) replicates relative to the ancestor, such that on average post-insemination success contributed 26.7% to 34.7% of total male reproductive success (Fig 2C). The BS-P&P and BS-PO regimes trended towards a higher fraction of total reproductive success that could be attributed to post-insemination success across replicate means. While not statistically significant, this trend potentially suggests that enhanced post-insemination competition positively affects fertilization success. Interestingly, post-insemination contribution increased to 79.7% in a single BS-P&P replicate. This evolutionary increase was due to a 13-fold increase in post-insemination success and only a 1.4-fold increase in total reproductive success.

Overall, the increased contribution of post-insemination dynamics was driven by the significant increase in post-insemination reproductive success of experimentally evolved males compared to ancestral males (Fig 2D; WS-P&P: z-value = 3.7, p < 0.001; BS-P&P: z-value = 3.6, p = 0.001; WS-PO: z-value = 3.4, p = 0.002; BS-PO: z-value = 4.0, p < 0.001). Once again, the BS-PO regime showed the strongest evolutionary response with a 6.8-fold increase from the ancestor, which supports the hypothesis that enhanced post-insemination competition increases the efficacy of sexual selection. Additionally, the WS-PO regime–the regime with the lowest levels of post-insemination competition–comparatively showed the lowest mean evolutionary change from the ancestor, though overall the evolutionary response was still strong. However, a post hoc test to determine if experimental evolution under directed sexual selection increased the rate at which post-insemination evolved relative to the WS-P&P baseline conditions showed no significant difference between regimes, suggesting a strong underlying selective pressure on sperm competitive ability.

Total reproductive success of experimentally evolved males compared to ancestral males also increased significantly across regimes (WS-P&P: z-value = 4.7, p < 0.001; BS-P&P: z-value = 2.7, p = 0.02; WS-PO: z-value = 3.5, p < 0.001; BS-PO: z-value = 3.6, p < 0.001), though to a lesser extent than post-insemination success alone (Fig 2D). Interestingly, only the BS-P&P regime showed a significant effect of sexual selection (z = -3.6, p < 0.001) compared to the baseline WS-P&P regime. Contrary to expectation [5,32], enhanced pre-insemination competition reduced the evolutionary response in male reproductive success. The WS-PO and BS-PO were not significantly different from the baseline. Thus, increasing the opportunity for pre-insemination sexual selection did not lead to faster evolution. Rather, enhanced pre-insemination competition appeared to hinder the rate of evolution of male reproductive success.

## Effective population size reflects strong selection

The effect population size ($N_e$) ranged from 9% to 16% of the census size (N = 5,000), depending on the estimator (Waples Plan II: 12% to 16%; Jonas Plan II: 9% to 11%), across all replicates and regimes (S4 Fig and S3 Data). Regimes where post-insemination interactions were isolated had on average lower effective population sizes across all chromosomes than the WS-P&P and BS-P&P regimes, regardless of the estimator. However, there was no significant effect of regime on $N_e$ (Waples Plan II Sampling: F = 0.98, DF = 3, p = 0.42; Jonas Plan II Sampling: F = 0.79, DF = 3, p = 0.51). Variance in reproductive success impacts $N_e$, especially when the sex ratio of breeding individuals is skewed. We calculated the upper bound on the number of breeding males [33], under the assumption that all females reproduced and the reduction in population size was due to variance in male reproductive success alone. For the estimated $N_e$ range, this analysis suggests that only 116–213 males reproduced (4.6–8.5% of the census male population), supporting strong sexual selection acted on males.

Given the XX/XO chromosomal sex determination system of *C. elegans*, we expected the estimated effective population size of the X chromosome to be approximately 75% of the estimated effective population size of the autosomes. However, the effective population size was not significantly different between the autosomes and sex chromosome (Waples Plan II: p = 0.3, mean $N_{e, auto}$ = 655.6, mean $N_{e, X}$ = 715.8; Jonas Plan II: p = 0.5, mean $N_{e, auto}$ = 498.2, mean $N_{e, X}$ = 478.3).

## Sperm competitive ability is a polygenic trait

We fit two complementary models to determine if the frequency of alleles at each SNP changed from the ancestral population to the experimentally evolved population (G31) in each regime.

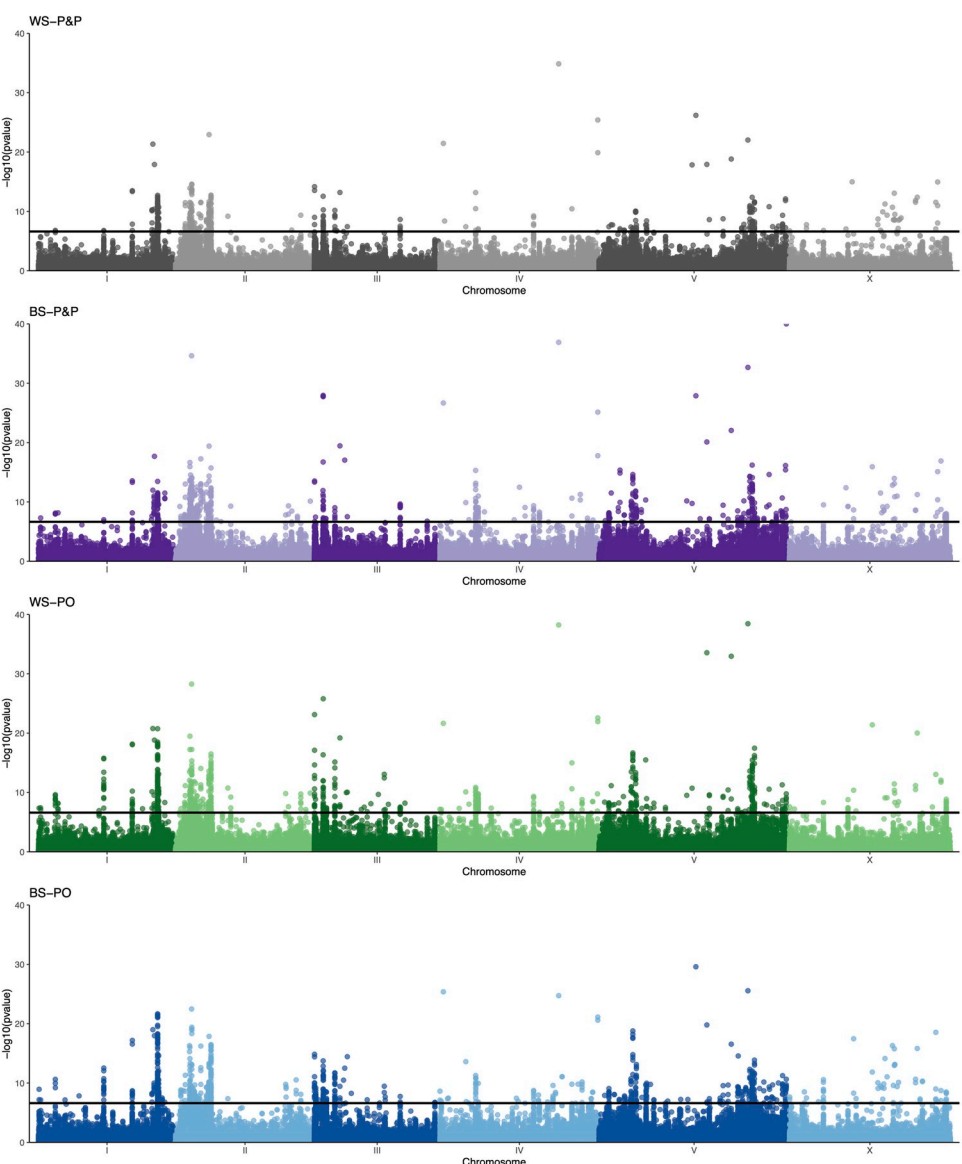

**Fig 3. Genomic response for each SNP over time fit for each regime (Model 2).** The horizontal line represents the Bonferroni significance threshold. Reproductive success is a highly polygenic trait with 40 peaks identified in the WS-P&P regime (gray), 71 in the BS-P&P regime (purple), 71 in the WS-PO regime (green), and 82 in the BS-PO regime (blue). The distribution of peak overlaps in shown in Fig 4.

Model 1 used a post hoc approach to compare SNP counts in the evolved and ancestral populations (Model 1: glm(SNP ~ regime), linear hypothesis test: Anc–Evolved$_{regime}$ = 0) and identified 1,819 significant SNPs after a Bonferroni correction (p < 1.86e-7). The significance trends of Model 1 (S4 Data) support the more robust findings of Model 2 (S5 Data). Here we fit independent models for each regime that included sampling at two intermediate generations (Model 2: glm(SNP$_{regime}$ ~ time)). In Model 2, we identified 127 non-overlapping significance peaks (i.e., at least three significant SNPs per 1kb window) across the five autosomes and the X chromosome, indicating that male reproductive success is polygenic (Fig 3 and S6 Data). Significance peaks showed a strong chromosome arm-center structure (Fig 3). While this pattern could be driven by a higher density of SNPs on the chromosome arms, we found more peaks

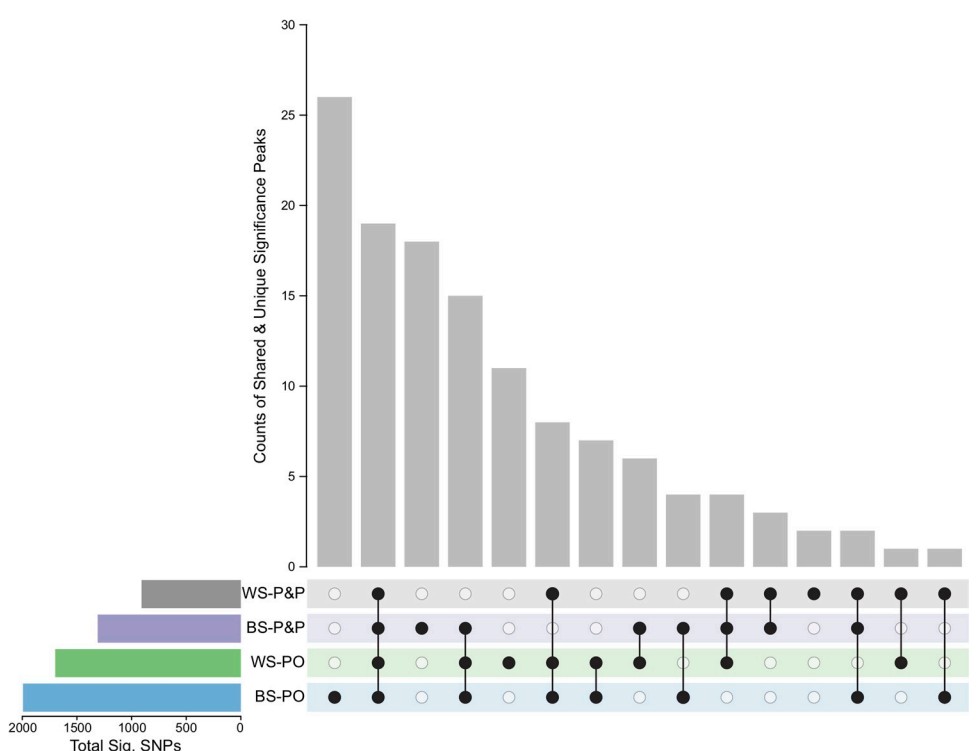

**Fig 4. Breakdown of significance peaks from Model 2.** The counts of significance peaks are shown along with the combination of regimes contributing to that count. Unique peaks are represented by a single black dot for the given regime. Shared peaks have multiple connected black dots. The total number of significant SNPs within each regime is given.

located on the chromosome arms than expected based on the ancestral SNP distribution ($\chi^2$ = 6.88, DF = 1, p < 0.01). One possible explanation for this deviation is that the higher recombination rate on the chromosome arms relative to the centers created beneficial haplotypes [29,30]. Nineteen peaks were shared across all regimes (Figs 3 and 4). The BS-PO regime had the highest number of significant SNPs (n = 1,994) as well as the highest number of unique significance peaks (n = 26). The BS-P&P regime represents the third highest grouping, reinforcing that gametic sexual selection resulted in a strong polygenic genomic response (Fig 4). The WS-P&P regime had the fewest number of significance peaks and only two peaks were unique to this regime. A conservative estimate of significance peaks (i.e., at least five significant SNPs per 1kb window) identified 77 non-overlapping peaks and recapitulated the pattern of enhanced gametic sexual selection driving a strong polygenic genomic response (S6 Data). Given the high degree of overlap, we present the analysis of the larger significance peak dataset.

Linkage disequilibrium was low between SNPs and significance peaks could be narrowed down to small genomic regions (S6 Data). The median peak width was 53 base pairs and the mean peak width was 453 base pairs. The largest peak spanned a 14,211 base pair region on the right arm of Chromosome I and lies in the intron of gene C17H1.2 (S5 Fig). This gene exhibits male-biased expression, though its function is uncharacterized [34]. The majority of significance peaks (n = 86) fell within a genic region, while 21 peaks were intergenic (S6 Data). Nineteen peaks were located in pseudogenes and one additional peak overlapped with a coding gene and pseudogene. Of all identified significance peaks, 79% (n = 100) were located in *C. elegans* hyper-divergent genomic regions [31].

To determine the functional pathways underlying improved male reproductive success, we examined the gene ontology (GO) terms associated with the genes underlying significance peaks (S6 Fig and S6 Data). The most common molecular function identified was SCF ubiquitin ligase complex formation through F-box proteins (n = 17). Several genes were also related to each carbohydrate binding, G-coupled protein receptor activity, and DNA binding. Six genes were associated with some form of RNA. However, 50% of genes were uncharacterized in function, identifying a lack of male-specific functional knowledge.

## Discussion

Quantifying the balance of pre- and post-insemination selection is critical for understanding how male reproductive fitness is comprised and how reproductive success evolves. This knowledge translates to better understanding how sexual selection contributes to population adaptation. We took a direct approach to isolate post-insemination from pre-insemination dynamics by coupling transgenic induction systems within an experimental evolution framework to examine whether these reproductive phases contribute in an additive or antagonistic manner to male reproductive fitness. All treatments showed a strong, rapid response to selection at both the phenotypic and genomic levels (Figs 2 and 3). Phenotypic results indicate that post-insemination selection was the major driver of male evolution. Genomic results support the importance of post-insemination selection and suggest that selection during this phase increased the efficacy of selection. Additionally, reproductive success is a highly polygenic trait with genes on all chromosomes contributing to the response to selection. These results provide new insights on the complexity of post-insemination dynamics and highlight the importance of considering all phases of reproduction.

The balance between pre- and post-insemination selection was complex and depended on the strength of selection imposed. At the phenotypic level, the within-strain competition treatments suggest that pre- and post-insemination act in an additive manner to increase male reproductive fitness (Fig 2). However, this pattern does not hold under enhanced between-strain competitive conditions. Instead, contrary to expectation, increased male-male competition (BS-P&P) decreased the rate of adaptation relative to base levels (WS-P&P). While this reduced adaptive rate under enhanced competitive conditions could be competitor-specific, it is more likely that this effect is wide-spread as experimental males were exposed to two different competitor strains. These increased competitive interactions could potentially harm females as a byproduct (i.e., sexual conflict) and therefore reduce female reproductive rate. However, the BS-PO treatment had the same number of males attempting to mate with females as the BS-P&P, the difference being that the BS-PO males could not transfer sperm post-mating. Thus, the increased number of males actually inseminating females is likely the contributing source of the decreased evolutionary response. While it seems possible that increased competition among sperm led to the decrease in fecundity [35], it is also possible that females altered egg-laying rates in response to the amount of sperm present as a result of a resource trade-off between reproductive and maintenance functions. To our knowledge, no studies have quantified this relationship in nematodes.

In contrast, increased sperm competition appeared to improve the rate of adaptation in males. BS-PO males trended towards the highest rate of increase in post-insemination success and post-insemination contributed the most to their overall reproductive response. While these comparative trends were not significant at the phenotypic level, at the genomic level populations evolved under increased sperm competition had the strongest genomic response across dozens of genes (Figs 3 and 4). This difference in sensitivity between the genotypic and phenotypic levels is likely due to the difference in power to detect small allele frequency

changes versus subtle changes that contribute to reproductive output. Interestingly, populations evolved under reduced sperm competitive dynamics (WS-PO) also showed a strong genomic response, suggesting that isolating post-insemination dynamics from pre-insemination dynamics allowed sexual selection to act more efficiently. While we isolated post-insemination through transgenic induction, this type of effect could be seen in nature if females were to mate with males over distinct periods of time and store sperm for later use.

Our method of population construction generated little haplotype structure, which allowed us to map genetic elements that responded to selection with high precision. A challenge in many quantitative trait loci [36] and evolve-and-resequence studies [37] is narrowing down the regions of selection to make specific statements on the genetic architecture of traits. In contrast, here we have high confidence that reproductive success and sperm competitive success are complex traits underlaid by over 60 genes (Figs 3 and 4). In most cases, we were able to narrow the region under selection to just a few hundred base pairs. While this precision should in principle allow us to identify the causal basis of the genetic response, given the highly polygenic structure of these complex traits, each contributing gene likely contributes a small effect, which makes the next step of functional molecular characterization challenging. To help prioritize this process, we performed a GO analysis to look for patterns in molecular functions or biological processes (S6 Fig). F-box proteins involved in protein-protein interactions, such as ubiquitin-ligase complex formation [38], showed a strong response in all treatments. Though their exact function is unknown, many of the several hundred *C. elegans* F-box genes show signatures of positive selection in wild isolates, suggesting that selective conditions observed in nature were mimicked in the lab [39]. However, nearly half of the identified genes were uncharacterized in function, despite *C. elegans* being a major model system. These genes represent a candidate list for future molecular studies to characterize the networks underlying male reproductive function. In particular, gene C17H1.2 is of interest for future study as it has a large significance peak falling within the second intron and exhibits male-biased expression patterns.

Sexual selection has a large effect on population size by limiting the number of successfully breeding adults, reviewed in [40]. We estimated the effective population size to be less than one fifth of the enforced census size. If one assumes that nearly all females are mated as an upper bound, this difference suggests that on average approximately 7% of males sired all offspring (S4 Fig). This is the very definition of opportunity for sexual selection [8] and is consistent with our conclusion that strong sexual selection acted on these populations even in the base level treatment (WS-P&P). Interestingly, the effective population size of the X chromosome was the same as the autosomes despite the XX/XO sex determination system of *Caenorhabditis* nematodes, which would suggest that the effective population size of the X chromosome should be 3/4 that of the autosomes under neutral expectations. This slight increase in the effective population size of the X chromosome may be further evidence of sexual selection, as the X chromosome is in males 1/3 of the time while autosomes are in males 1/2 the time, and so the autosomes are more susceptible to drift induced by variance in mating success specially among males [40,41]. The X chromosome also had the fewest number of significance peaks, so in addition to the demography of the X chromosome itself, it is also possible that there may be additional reductions in autosomal variation due hitchhiking [40].

Darwin first noted that the existence of elaborate sex-specific traits seemed at odds with regular evolutionary processes, and more than a hundred of years of research has subsequently focused on understanding how sexual selection drives diversity for these traits within and between populations. Our work indicates that the cryptic phenotypes and molecular effects that emerge during post-insemination interactions are equally important in determining fertilization success and likely to be just as genetically complex.

## Materials and methods

### Molecular biology

Guides targeting sequences in the same intergenic regions utilized by the ttTi4348 and ttTi5605 MosSCI sites have been previously described [21,42]. Additional guide sequences were chosen using the Benchling CRISPR design tool, based on the models of Doench *et al.* [43] and Hsu *et al.* [44], and the Sequence Scan for CRISPR tool [45]. Guides were inserted into pDD162 (Addgene #47549) [46] using the Q5 site-directed mutagenesis kit (NEB) or ordered as cr:tracrRNAs from Synthego. A complete list of guide sequences can be found in S1 Table.

Repair template plasmids were assembled using the NEBuilder HiFi Kit (NEB) from a combination of restriction digest fragments and PCR products. PCR products were generated using the 2x Q5 PCR Master Mix (NEB) in accordance with manufacturer instructions. Details of plasmid construction can be found in the supplemental methods and S2 and S3 Tables. Plasmids were purified using the ZR Plasmid Miniprep kit (Zymo) and all plasmid assembly junctions were confirmed by Sanger sequencing.

### Strain generation

All strains used in this study are listed in S4 Table and depicted schematically in S2 Fig. Insertion of transgenes was done by CRISPR/Cas9 using standard methods. Briefly, a mixture of 10ng/µl repair template plasmid, 50ng/µl plasmid encoding CAS9 and the guide RNA and 2.5ng/µl pCFJ421 (Addgene #34876) [47] was injected into the gonad of young adult hermaphrodites. Where hygromycin resistance (HygR) was used as a selectable event, two to three days after injection, hygromycin B (A.G. Scientific, Inc.) was added to the plates at a final concentration of 250µg/ml. Successful insertion was confirmed by PCR and Sanger sequencing.

To generate the male sterility induction strain PX624, *pie-1p*::TIR-1 was inserted into the Chromosome I site and a degron tag was added to the native *spe-44* locus of JU2526 as in Kasimatis *et al.* [21] (S2A and S2C Fig). The majority of exons 2–4 of the native *fog-2* gene were then deleted using the guides and oligonucleotide repair template listed in S1 and S3 Tables. Microinjections and *dpy-10* co-marker screening were done as previously described [21,48]. The degron tag on *spe-44* causes a slight decrease in total fecundity [21], however all subsequent experimental evolution treatments contain this genetic manipulation and therefore it does not affect the interpretation of our results. This strain represents the predecessor for the experimental evolution ancestral population (see "Generating genetic diversity").

The *hsp-16.41p*::PEEL-1 + *rpl-28p*::mKate2 + *rps-0p*::HygR three gene cassette was inserted into the Chromosome I site of CB4856. Individuals with confirmed inserts were crossed to JK574, containing *fog-2*(q71), and backcrossed 4 times to CB4856 (S2B Fig). A single pair was then chosen for 14 generations of inbreeding to create strain PX626. To introduce a second copy of *hsp-16.41p*::PEEL-1, a *hsp-16.41p*::PEEL-1 +*loxP*::*rps-0p*::HygR::*loxP* two gene cassette was inserted into the Chromosome II site (S2 Fig). The HygR gene was then removed by injection of a CRE expressing plasmid pZCS23 [49] at 10ng/µl, with removal monitored by PCR, to generate PX630. PX626 was crossed to PX630 to generate the final novel, bioassay competitor strain PX631 (S2E Fig).

To generate a lethality and male sterility induction strain, PX624 was crossed with PX631 and then backcrossed 5 times with PX624 to introgress *hsp-16.41p*::PEEL-1 in the Chromosome II site to create strain PX655. Since the Chromosome I site of PX624 is occupied by *pie-1p*::TIR-1, CRISPR/Cas9 was used to insert the *hsp-16.41p*::PEEL-1 + *rpl-28p*::mKate2 + *rps-0p*::HygR three gene cassette into PX624 at a site on Chromosome III between *nac-3* and

K08E5.5 that has not been previously used for transgene insertion, creating PX656. PX655 and PX656 were then crossed to create the final competitor strain PX658 (S2D Fig).

## Generating genetic diversity

The male sterility induction strain (PX624) was exposed to ethyl methanesulfonate (EMS) to induce genetic variation (S2 Fig). Populations of 8,000–10,000 age-synchronized L4 worms were divided into 4 technical replicates and suspended in M9 buffer. Worms were incubated in 12.5 mM EMS for 4 hours at 20˚C, after which they were rinsed in M9 buffer and plated on NGM-agar plates. Replicate populations were given two recovery and growth generations with ample food following a mutagenesis event. A total of five low-dose mutagenesis rounds coupled with recovery generations were performed. During each of the recovery rounds, a subset of worms from each replicate were screened on NGM-agar plates containing 1 mM indole-3-acetic acid (Auxin, Alfa Aesar) following Kasimatis et al. [21] to test if mutagenesis had compromised the integrity of the sterility induction system. Specifically, if eggs were observed on an auxin-containing plate, then that replicate was removed and another replicate was subdivided, so a total of four replicate populations were always maintained.

After the final round of mutagenesis and recovery, replicate populations were maintained for five generations of lab adaptation. They were then combined for an additional 10 generations of lab adaptation with a population size of approximately 30,000 worms. The integrity of the sterility induction system continued to be screened every two generations throughout the entire lab adaptation process. This genetically diverse, male sterility induction strain PX632 represents the ancestral experimental evolution population (S2 Fig).

## Experimental design and worm culture

The ancestral population (PX632) was divided into four experimental regimes, which varied based on total (i.e., pre- and post-insemination) or sperm (i.e., post-insemination) competition dynamics occurring either within the evolving strain alone or between the evolving strain and competitor strain (PX658): within-strain pre- and post-insemination competition (WS-P&P), within-strain post-insemination only competition (WS-PO), between-strain pre- and post-insemination competition (BS-P&P), and between-strain post-insemination only competition (BS-PO).

Each regime had six replicate populations evolved for 30 generations. Ten selective events occurred over the course of experimental evolution denoted by the induction of sterility, the addition of competitors, and the induction of sterility and addition of competitors in the WS-PO, BS-P&P, and BS-PO regimes, respectively (Figs 1, 2A and S1). The WS-P&P had no direct selection applied. Each selective event was followed by a recovery generation, where no direct selection was applied, to allow the populations to return to the census size. During the recovery generation, a subset of worms from the regimes with sterility induction were screened on auxin-containing plates to ensure the sterility induction system was functional. Additionally, a subset of worms from all replicates was frozen for potential future research. The detailed selection procedure follows and is depicted schematically in S1 Fig.

To start each selective event age synchronized L1 worms were plated onto five 10 cm NGM-agar plates seeded with OP50 *Escherichia coli* at 20˚C with a density of 1,000 worms per plate, giving a census size of 5,000 worms per replicate per regime [50,51]. Forty-eight hours later, experimental regimes with sterility induction (WS-PO and BS-PO) were transferred to NGM-agar plates containing 1mM auxin. Experimental regimes without sterility induction (WS-P&P and BS-P&P) were transferred to fresh NGM-agar plates. For all transfers, worms within a replicate were pooled and then redistributed across five plates with a density of 1,000

worms per plate. After 24 hours, males from the competitor strain PX658 were filter-separated from females using a 35 um Nitex nylon filter and added to experimental regimes with competition at a mean density of 200 competitor males per plate (evolving to competitor ratio of 1:2.5). After another 24 hours, eggs were collected from all replicates, hatched, and age synchronized. To ensure that only progeny from the evolving males and not from the competitor males were being propagated, larval lethality of competitor progeny was induced following Seidel *et al.* [24]. Briefly, approximately 5,000 L3 worms were suspended in 5 mL of S-Basal and heat-shocked in a 35˚C sealed water bath for 2.5 hours to activate ectopic expression of the lethal protein PEEL-1. After heat-shock, worms were plated on NGM-agar plates to end the selective event. All experimental regimes were subjected to the heat-shock procedure, even if competitor worms were not added.

A subset of approximately 200 worms from the competition and sterility and competition regimes were removed prior to heat-shock and fluorescence screened to determine the proportion of progeny coming from the competitor worms, which expressed red fluorescent protein (RFP), versus the evolving worms, which had no fluorescence.

The competitor strain PX658 was maintained on NGM-agar plates seeded with OP50 *E. coli* at 20˚C in population sizes of approximately 20,000 worms. The competitor strain was reset from freezer stocks every 3 weeks (~4 generations) to prevent adaptation and maintain a constant competitive phenotype.

## Fertility assays

We assayed the fertility of the ancestor and all the evolved replicates (N = 13 populations) to determine the total competitive reproductive success of males as well as their sperm competitive success. The assay conditions followed the same experimental timeline for the induction of sterility and addition of competitors as the experimental evolution regime. Total competitive reproductive success was assessed by adding the novel competitor PX631 in equal proportion to evolving males. Sperm competitive success was assessed by inducing sterility of the evolving male before adding the novel competitor in equal proportion to evolving males. The use of the novel competitor and high competition ratio acted as a "stress-test" of male competitive ability. Both assays were performed with a population of 250 evolving females, 250 evolving males, and 250 novel competitors. After a 24-hour competition period, eggs were collected, hatched, and age synchronized for screening. At least 200 L3 progeny were counted for each assay and then fluorescence-screened for the proportion of progeny coming from evolving (RFP minus) or competitor (RFP plus) males. Three independent biological replicates were done for each assay across all experimental evolution replicates (S7 Data).

Fertility data were analyzed using the R statistical language v4.0.0 [52]. An equality of proportions test was performed on the ancestral data to determine if ancestral males sired half the total progeny under total competitive and sperm competitive conditions. The evolved male fertility data were analyzed using a linear model (GLM) framework with random effects using the *lme4* v.1.13 package [53]. The *multcomp* package [54] was then used to perform a planned comparisons tests with defined contrasts to determine if: i) evolutionary change from the ancestral population occurred, and ii) experimental evolution under direct sexual selection affected reproductive success differently than baseline selection alone (i.e., WS-P&P).

## Genome sequencing, mapping, and SNP calling

We performed whole-genome sequencing on pooled samples of 2,000–3,000 L1 worms from the ancestral population (G0) and each replicate per regime at generations 13, 22, and 31. Three independent pooled extractions were done for the ancestral population (i.e., generation

0) to capture as many segregating variants as possible. Worms were flash frozen and DNA was isolated using Genomic DNA Clean and Concentrator-10 (Zymo). Libraries were prepared using the Nextera DNA Sample Prep kit (Illumina) starting from 5 ng of DNA. 100 bp paired-end reads were sequenced on an Illumina HiSeq 4000 at the University of Oregon Genomics and Cell Characterization Core Facility (Eugene, OR). The average genome-wide sequencing coverage for generations 0, 13, 22, and 31 was 162×, 24×, 26×, 50×, respectively.

Reads were trimmed using skewer v0.2.2 [55] to remove low quality bases (parameters: -x CTGTCTCTTATA -t 12 -l 30 -r 0.01 -d 0.01 -q 20). The trimmed reads were mapped to the *C. elegans* N2 reference genome (PRJNA13758-WS274) [34] using BWA-MEM v0.7.17 (parameters: -t 8 -M) [56] and then sorted using SAMtools v1.5 [57]. We removed PCR duplicates with MarkDuplicates in Picard v2.6.0 (https://github.com/broadinstitute/picard), realigned insertions/deletions with IndelRealigner in GATK v3.7 (https://github.com/broadinstitute/gatk/#authors), and called variants with mpileup in bcftools v1.5 [58]. The mpileup file was then converted to a genotype-called vcf file, insertions/deletions were removed, and the allelic depth was extracted for all diallelic SNPs for further analysis.

To improve the reliability of the analysis pipeline, additional filtering was done using R [52]. Repeat regions were removed first by generating the ranges of repeat regions (https://gist.github.com/danielecook/cfaa5c359d99bcad3200) based the *C. elegans* N2 masked reference and then filtering them out in R. SNPs in the upper and lower 5% tails of the total coverage distribution (i.e., >342× and ≤20×, respectively) were removed. This yielded a total of 321,929 SNPs to be considered for analyses.

### Estimation and candidate SNP inference

Genetic diversity summary statistics were estimated for the ancestral population. Coverage-weighted average heterozygosity ($\pi$) was calculated following Begun et al. [59]. SNP density ($\theta_w$) was calculated across 1kb sliding windows. We performed a Kolmogrov-Smirnov test to determine if the site frequency spectrum, $\pi$, and $\theta_w$ differed between chromosome arm domains and center domains [29]. Effective population size ($N_e$) was calculated per chromosome using a common set of SNPs for each of the evolved regime replicates following both the Waples [60] Plan II sampling and Jonas [61] Plan II sampling using the R package poolSeq [61]. An analysis of variance was performed in R to determine if the genome-wide $N_e$ differed between regimes and Welch's Two-Sample t-test was performed to determine if the estimated $N_e$ on autosomes differed from the X chromosome. We estimated the upper bound on the number of breeding males ($N_m$) by solving the equation $N_e = (4 N_m N_f) / (N_m + N_f)$ for $N_m$ using the estimated effective population sizes and assuming that all females reproduced ($N_f = 2,500$).

Allele count data were analyzed using R [52] following two complementary models. Model 1 fit allele counts for ancestral and evolved populations using a generalized linear mixed model with a binomial logistic distribution: glm(SNP ~ regime). The SNP data going into Model 1 were filtered to ensure each SNP was present in the ancestor and at least ten of the evolved samples (i.e., regime x replicate). A total of 263,373 SNPs fit the full model (S4 Data). The *multcomp* package [54] was then used to perform a planned comparisons tests with defined contrasts to determine if experimental evolution under direct sexual selection affects the genome differently than baseline selection alone (i.e., WS-P&P). Model 2 fit allele counts across all time points for each regime separately, again using a generalized linear mixed model with a binomial logistic distribution: glm($SNP_{regime}$ ~ time). The SNP data going into Model 2 were filtered to ensure each SNP was present in the ancestor and at least nine occurrences across replicates and time points. A total of 202,926 SNPs, 222,731 SNPs, 200,324 SNPs, and 204,946

SNPs fit the full model for the WS-P&P, WS-PO, BS-P&P, and BS-PO regimes, respectively (S5 Data). We compared the results of Model 2 to the same model fit with a quasi-binomial distribution. The quasi-binomial distribution was hypersensitive to outlier samples, and therefore we present the results of the binomial distribution. For both Models 1 and 2, significance was determined using a genome-wide Bonferroni cut-off.

A significance peak was called if three or more significant SNPs fell in a 1kb window. Peaks were classified as occurring within a gene (intragenic) or between genes (intergenic) using JBrowse in WormBase [34]. If multiple 1kb windows fell within a single gene, then the windows were combined and called as a single intragenic peak. For a more conservative estimate, we repeated this analysis using a threshold of five or more significant SNPs within a 1kb window. The molecular and biological functions of the associated genes were determined using gene ontology analysis in UniProt [62] and QuickGO [63].

## Supporting information

**S1 Fig. Selective event timeline for each regime.** A selective event was initiated by plating age-synchronized larval stage 1 (L1) worms on regular nematode-growth media (shown in gray). Throughout larval development (L1-L4) worms were not reproductively active. However, spermatogenesis began in L4 males (shown by the crossed-out boxes) and therefore sperm was produced and stored prior to mating. Day 1 (D1) of adulthood corresponds to day 1 of the experimental timeline (see Fig 1). The WS-PO and BS-PO regimes were transferred to auxin-containing media (shown in green), while the WS-P&P and BS-P&P regimes remained on regular media. All males, regardless of regime, were fully fertile at the start of day 1. Females began laying eggs on day 1. On day 2 of adulthood, males on auxin-containing media (i.e., WS-PO and BS-PO) were fully sterile and could no longer transfer sperm. Competitor males were added to the BS-P&P and BS-PO regimes on day 2. Females laid eggs throughout adulthood, however, only eggs laid in the last 6–8 hours of day 2 had a thick enough eggshell to survive the propagation process. On day 3, populations were propagated by bleach-killing larvae and adults and then age-synchronizing surviving eggs. The resulting L1s were again plated on regular media and grown to L3 before heat-shocking to kill progeny coming from the competitor males.
(TIFF)

**S2 Fig. Schematic of strain construction. A)** The components for creating an obligate outcrossing sterility induction line were genetically engineered in the wild isolate background JU2526. The spermatogenesis gene *spe-44* was degron-tagged and TIR1 was inserted to create strain PX737. The hermaphrodite self-sperm gene (*fog-2*) was knocked-out to create strain PX738. These strains are used in panels C and D. **B)** To generate an inducible lethality line, heat-shock driven *peel-1* was inserted into the CB4856 background on Chromosomes I and II to create strains PX739 and PX630, respectively. These strains are used in panels D and E. **C)** Strains PX737 and PX738 were crossed to creating a male-female, inducible sterility triple mutant (PX624). Strain PX624 went through five low dose rounds of mutagenesis each followed by two recovery generations. After the final recovery generation, the population was expanded for 15 generations of lab adaptation to create the experimental evolution ancestral population (PX632). **D)** The competition strain has five transgenic modifications. Heat-shock driven *peel-1* was inserted on Chromosome III of strain PX737, creating an inducible lethality and inducible sterility strain (PX656). Strains PX624 and PX631 (panel E) were crossed to given another inducible lethality and sterility double mutant. These worms were backcrossed to PX624 five times to give a predominantly JU2526 genomic background. This strain, PX655, was crossed with PX656 yielding a quintuple mutant, which was inbred to three generations

followed by five generations of lab adaptation. The final strain PX658 served as the competitor during experimental evolution. **E)** A separate bioassay competitor strain was generated by introgressing the *fog-2*(q71) mutation into PX739. These worms were backcrossed to the CB4856 genomic background four times and then inbred for 14 generations, creating strain PX626. This strain was crossed to PX630 to create an obligate outcrossing strain with two heat-shock driven *peel-1* insertions. The final strain PX631 served as the novel competitor during phenotypic assays.
(TIFF)

**S3 Fig. Genetic diversity of the ancestral population. A)** The minor allele frequency (MAF) across Chromosome II (as an exemplar). The genome-wide mean is shown in blue. **B)** Histogram of MAF counts across the entire genome binned by chromosome arms and chromosome center. Values of zero are excluded from the plot. **C)** Nucleotide diversity ($\pi$) calculated per SNP across Chromosome II. The genome-wide mean is shown in blue. **D)** Histogram of nucleotide diversity across the entire genome binned by chromosome arms and chromosome center. Values of zero are excluded from the plot. **E)** SNP density ($\theta_w$) per base pair across Chromosome II. The genome-wide mean is shown in blue. **F)** Histogram of SNP density in 1kb windows across the entire genome binned by chromosome arms and chromosome center.
(TIFF)

**S4 Fig. The estimated effective population size ($N_e$) per chromosome for all replicates.** The effective population size was greatly reduced compared to the census size (N = 5,000). Regime did not have a significant effect on effective population size (Waples Plan II Sampling [60]: F = 0.98, DF = 3, p = 0.42; Jonas Plan II Sampling [61]: F = 0.79, DF = 3, p = 0.51).
(TIFF)

**S5 Fig. Zoom plot of the major significance peak on the right arm of Chromosome I.** Significant SNPs pile up in the second intron of gene C17H1.2. This gene has male-biased expression, though it's function is uncharacterized.
(TIFF)

**S6 Fig. The molecular functions for genes associated with significance peaks based on a GO analysis.** Ubiquitin ligase complex formation through F-box proteins, carbohydrate binding, G-coupled protein receptor activity, and DNA binding were the most common functions identified. However, the majority of genes are yet uncharacterized in function.
(TIFF)

**S1 Table. Guide sequences.** The guide sequence, genomic location, target region/gene, and format (plasmid or cr:tracrRNA) are given.
(XLSX)

**S2 Table. Plasmid construction.** The plasmid name and insert are given for both plasmids used in construction and as repair templates.
(XLSX)

**S3 Table. Primers used in this study.** The primer name, sequence (in 5' to 3' orientation), and purpose for a given primer are listed.
(XLSX)

**S4 Table. Strains generated in this study.** Full genotype information for each strain used in this study, along with the genomic background, method of construction, and generations of backcrossing and/or inbreeding.
(XLSX)

**S1 Data. SNP data for the ancestor.** The chromosome, position (in base pairs), reference allele, alternate allele, counts of reference alleles, counts of alternate alleles, total coverage, minor allele frequency (MAF), chromosome domain, and nucleotide diversity ($\pi$) are given. (TXT)

**S2 Data. Watterson's theta calculated in 1kb windows across each chromosome.** The chromosome, chromosome domain, theta per window, and theta per base pair are given. (TXT)

**S3 Data. Effective population size estimated using Waples Plan II sampling and Jonas et al. Plan II sampling for each replicate and each chromosome.** (TXT)

**S4 Data. Summary statistics for the Model 1 planned comparison analysis of ancestral versus evolved allele counts.** For each SNP, the chromosome and position (in base pairs) is given along with the slope estimate and p-value for each regime comparison. (ZIP)

**S5 Data. Summary statistics for the Model 2 GLM analysis of allele counts over time for each regime.** For each SNP within each regime, the chromosome and position (in base pairs) is given along with the model intercept, slope estimate, standard error, z-value, and p-value. (TAR.GZ)

**S6 Data. Summary of the significance peaks identified using the Model 2 genomic results.** The chromosome, start position (in base pairs), stop position (in base pairs), presence in each treatment, associated gene, genetic region, molecular function (from GO analysis), and biological function (from GO analysis) are given. Significance peaks were determined based on a criteria of at least three significant SNPs per 1kb window or a conservative estimate of at least five significant SNPs per 1kb window. (XLSX)

**S7 Data. Competitive phenotyping data for the ancestor and all evolved replicates.** (XLSX)

## Acknowledgments

We thank Brennen Jamison, Erik Johnson, and Christine Sedore for assistance during experimental evolution and Anastasia Teterina for advice on the genomic analyses. We thank Levi Morran, Bill Rice, Locke Rowe, and the Phillips lab for their helpful discussion. This work was conducted in part using the resources of the University of Oregon Genomics and Cell Characterization Core Facility and Research Advanced Computing Services.

## Author Contributions

**Conceptualization:** Katja R. Kasimatis, Patrick C. Phillips.

**Data curation:** Katja R. Kasimatis.

**Formal analysis:** Katja R. Kasimatis.

**Funding acquisition:** Patrick C. Phillips.

**Investigation:** Katja R. Kasimatis, Megan J. Moerdyk-Schauwecker, Ruben Lancaster, Alexander Smith, John H. Willis.

**Methodology:** Katja R. Kasimatis, Megan J. Moerdyk-Schauwecker.

**Software:** Katja R. Kasimatis.

**Supervision:** Patrick C. Phillips.

**Validation:** Patrick C. Phillips.

**Visualization:** Katja R. Kasimatis.

**Writing – original draft:** Katja R. Kasimatis, Megan J. Moerdyk-Schauwecker, Patrick C. Phillips.

**Writing – review & editing:** Katja R. Kasimatis, Megan J. Moerdyk-Schauwecker, Ruben Lancaster, Alexander Smith, John H. Willis, Patrick C. Phillips.

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
