## [Decision Letter · Decision Letter 0]

18 Aug 2021

Dear Dr Kasimatis,

Thank you very much for submitting your Research Article entitled 'Post-Insemination Selection Dominates Pre-Insemination Selection in Driving Rapid Evolution of Male Competitive Ability' to PLOS Genetics.

The manuscript was fully evaluated at the editorial level and by independent peer reviewers. The reviewers appreciated the attention to an important problem, but raised some substantial concerns about the current manuscript. Based on the reviews, we will not be able to accept this version of the manuscript, but we would be willing to review a revised version. We cannot, of course, promise publication at that time.  In the revision, please ensure that the manuscript is framed for and accessible to a broad genetics audience.  In addition, there are several questions about the experimental evolution protocol that need to be addressed, some of which include acknowledging the limitations of the design.  Finally, the reviewers have many helpful suggestions about the Ne and pool-seq analyses that can make these analyses more robust. 

Should you decide to revise the manuscript for further consideration here, your revisions should address the specific points made by each reviewer. We will also require a detailed list of your responses to the review comments and a description of the changes you have made in the manuscript. Please note that two of the reviewers include substantial comments in attached pdfs, and that these must also be addressed in the same way.

If you decide to revise the manuscript for further consideration at PLOS Genetics, please aim to resubmit within the next 60 days, unless it will take extra time to address the concerns of the reviewers, in which case we would appreciate an expected resubmission date by email to plosgenetics@plos.org.

[LINK]

We are sorry that we cannot be more positive about your manuscript at this stage. Please do not hesitate to contact us if you have any concerns or questions.

Yours sincerely,

Kelly A. Dyer

Associate Editor

PLOS Genetics

Bret Payseur

Section Editor: Evolution

PLOS Genetics

Reviewer's Responses to Questions

**Comments to the Authors:**

Reviewer #1: This is a really nice paper reporting interesting results from a very elegantly designed experiment. To my knowledge, this is the first careful dissection/quantitation of pre- and post-insemination selection and their relative efficacy in modulating overall reproductive success. Thus, the results are of non-trivial interest to those working on sexual selection and sexual conflict. Moreover, there is a lot of genomic information that points to future work in identifying important genes in the response to pre- and post-insemination selection. I have no substantive critiques; a few small comments/queries to authors are noted on the uploaded manuscript as pop-up notes.

Reviewer #2: “Post-insemination selection dominates pre-insemination selection in driving rapid evolution of male competitive ability” utilizes the advantages of C. elegans to uniquely parse apart sexual selection on pre and post insemination. The experimental design is fully factorial to test total or post-insemination sexual selection when competition occurs either within or between populations. After 10 selection cycles spread across 30 generations the authors show differences in competitive insemination success against a novel competitor and sequencing of the different populations indicates that the this is polygenic with many peaks associated with this trait. Utilizing the genetic tools in C. elegans appears to be a very creative approach to tease apartment these questions, but I remain stuck on an aspect of the experimental design that makes it difficult for me to interpret the subsequent results.

Major Concerns:

The authors compare total (pre + post insemination selection) to only post-insemination selection. From my understanding, the pre-insemination competition is eliminated by sterilizing the worms using an Auxin induced system to stop sperm production. This experiment occurs on Day 1, so does it mean that females are only inseminated with sperm from worms that are the first to copulate, before the auxin begins to take effect? If this is the case, it seems like there is still pre insemination competition and there is selection for fastest to copulate. I didn’t see evidence for this, but wondered if perhaps the SPE-44 degron at an intermediate frequency in the population so only some of the worms are sterilized? The graphical images (figure 1, figure 2A) make it seem like the number of inseminated sperm decreases in this auxin treatment, but that does not seem congruent with the method described. It’s possible that I’m missing something, but I have spent a lot of time rereading the methods and staring at the graphical cartoons and I can’t fully understand what this step was selecting for, which makes it difficult to interpret if it is truly representing pre-insemination selection.

The introduction presents pre-insemination selection as mating success and it is unclear to me how sterilizing the worms reduces mating success. From what I understand about the system the worms may still “compete” for the female copulate and probably also copulate with them, just not successfully transfer sperm. My interpretation is that they are still reducing pre-insemination mating success while sterile, as they reduce the time non-sterile males may copulate and transfer their sperm.

For the BS-PO experiment, does the temporal order of copulation effect the sperm usage? If the order of copulation plays a role in the sperm usage, it seems like this may not be selecting for post-insemination sperm success and instead just insemination order.

Please clarify if the ancestral males were pre or post EMS treatment. If the ancestral population has the genetic diversity introduced by the EMS and WS-P&P population is not undergoing selection, what is the hypothesis for the large difference in post-insemination success?

Is there a possibility that female choice is also evolving in these populations?

I think supplemental figure 4 that shows shared and unique peaks is interesting and should be considered including in the primary text.

Does the sequencing of the G0s indicate if all of the significant SNPs exist in all of the founding populations, or is there the possibility of founders effect?

Minor Concerns:

- Throughout the paper the authors refer to “evolving males”; this implies that there must be evolution occurring in this selection experiment. I would feel more comfortable if they were referred to as “experimental males”, as this leaves it open to the results if all of the treatments resulted in evolution.

- Line 32-33: “we find that after 30 generations…” implies that selection occurred over all 30 generations, not the 10 selective rounds over 30 generations.

- Line 43: not sure what is meant by “between the sexes”. Please clarify.

- Line 183-186: “The BS-P&P and WS-PO regimes trended towards higher fraction of total reproductive success that could be attributed to post-insemination success across replicate means…” I think this references Figure 2C, but do not see this pattern in the data. Do you mean BS-P&Pand BS-PO? This trend is small, are there any statistics to back it up?

- Have the significance levels in the fold-change comparisons (Figure 2D) been adjusted for multiple comparisons?

- Adding the treatment as a title to each Manhattan plot in figure 3 would help comparing them easier.

- Line 456: “The assay conditions mimicked the environment under which the worms evolved.” I am unsure of what this means. My initial reading was each strain only received the treatments for which they had been selected (P&P vs PO and WS vs BS). But surely that can’t be correct, as the within-strain experimental treatments that never experienced competition from another strain must have also been introduced to the novel competitor. Does it mean that only the WS-PO and BS-PO were exposed to the auxin before the competition experiment? If that is the case, are all the experimental worms sterile? Figure 2D suggests that all of the experimental conditions experienced both treatments.

Reviewer #3: Kasamitis et al. apply a novel transgenic system to decouple pre- and post-copulatory competition in C. elegans. Factorial experimental evolution from induced mutations is followed by competitive fitness assays and pooled sequencing.

The main findings are:

strong responses to selection across all four regimes, particularly in post-insemination competition

evolution was slowed by pre-insemination competition

the response to selection was polygenic, and most genes implicated are of unknown function

evolution of the X chromosome differed from autosomes

This work addresses an important question with a new, powerful method to isolate components of male fitness.

Although large differences in response across regimes were not seen, the results obtained and utility of the method are noteworthy advances, in particular for our understanding of male C. elegans biology.

I have some minor issues with framing, and some technical issues that need to be addressed (which could possibly change the results a little, but not much I think), detailed below. Other questions/suggestions are commented in the pdf.

Framing:

In general, the paper is written with a lot of worm biology assumed, such as reproductive schedules. Please try and flesh this out in the introduction and methods to make the paper more accessible to others. For example, when is sperm production happening in males, at the relevant points during experimental evolution?

The suggestion that post-insemination dynamics are neglected seems a bit overdone, given the enormous body of literature on the importance of gametic interactions (e.g. in free-spawning animals). Similarly, for organisms without flashy phenotypes (e.g. worms), there is of course ample precedent for the predominance of evolution in "the unseen world of molecular interactions".

Technical issues requiring attention:

In analysis of Ne, the union of called SNPs shared with the ancestor are used for each individual sample, which is therefore confounded with sequencing depth. It would be better to use a set of common markers for all estimates (r^2 for Ne~nSNPs is >0.25 at G31).

Secondly, the PlanII method of Waples is used, which has been shown to be upwardly biased due to neglecting additional sampling associated with poolseq data (Jonas et al, 2016). A quick reanalysis using some of the estimators from Jonas et al. (R package poolSeq) shows lower values (mean ~420-520 at G31 across the four regimes at a set of ~35k common [called in >63/73 samples] markers), and the X chromosome is similar to autosomes (still unexpected perhaps, but less so than a significantly larger Ne. Note that Jonas et al. also saw similar X/auto values in a reanalysis of some fly data).

The authors have, reasonably, used only the terminal G31 sample for Ne estimation where drift variance is maximal. Looking all time points, at the chromosome level, using a set of common markers across all samples, there is some evidence for an effect of regime: Ne appears to have flatlined for the two between-strain competition, but not for within-strain competition regimes. Make of this what you will.

>20% of the genome is repeat masked (proportionally much less for the X), but diversity stats do not seem to be adjusted for this.

There looks to be an error in the calculation for theta (line 123) - the window used is 10000, not 1000 bp.

In analysis of the poolseq data, the Bonferroni cutoff is set as 1/(0.5*N), rather than alpha/N.

I also ask in the pdf to compare results from a quasibinomial glm - binomial error can be unrealistic for poolseq data leading to a high false positive rate, though I suspect the difference will not be large for your analyses.

Question:

Is anything happening on the mitochondrion?

**Have all data underlying the figures and results presented in the manuscript been provided?**

Reviewer #1: Yes

Reviewer #2: Yes

Reviewer #3: Yes

PLOS authors have the option to publish the peer review history of their article (what does this mean?). If published, this will include your full peer review and any attached files.

Reviewer #1: No

Reviewer #2: No

Reviewer #3: No

---

## [Decision Letter · Decision Letter 1]

28 Jan 2022

Dear Dr Kasimatis,

We are pleased to inform you that your manuscript entitled "Post-Insemination Selection Dominates Pre-Insemination Selection in Driving Rapid Evolution of Male Competitive Ability" has been editorially accepted for publication in PLOS Genetics. Congratulations! 

Yours sincerely,

Kelly A. Dyer

Associate Editor

PLOS Genetics

Bret Payseur

Section Editor: Evolution

PLOS Genetics

Comments from the reviewers (if applicable):

Reviewer's Responses to Questions

**Comments to the Authors:**

Reviewer #1: When I reviewed the original submitted version, I liked the study and most of my comments were seeking clarification, occasionally some further discussion. I had no substantive critique and I felt that the study and its results are of interest to people in evolutionary genetics. The revised version has addressed all my comments most satisfactorily, and now is a much clearer manuscript in many respects (thanks also to the author's responding to queries from other reviewers). I commend the authors for a thorough and serious revision.

**Have all data underlying the figures and results presented in the manuscript been provided?**

Reviewer #1: Yes

PLOS authors have the option to publish the peer review history of their article (what does this mean?). If published, this will include your full peer review and any attached files.

Reviewer #1: No

**Data Deposition**

http://datadryad.org/submit?journalID=pgenetics&manu=PGENETICS-D-21-00891R1

**Press Queries**

---

## [Editor Report · Acceptance letter]

9 Feb 2022

PGENETICS-D-21-00891R1 

Post-Insemination Selection Dominates Pre-Insemination Selection in Driving Rapid Evolution of Male Competitive Ability 

Dear Dr Kasimatis, 

We are pleased to inform you that your manuscript entitled "Post-Insemination Selection Dominates Pre-Insemination Selection in Driving Rapid Evolution of Male Competitive Ability" has been formally accepted for publication in PLOS Genetics! Your manuscript is now with our production department and you will be notified of the publication date in due course.

With kind regards,

Olena Szabo

PLOS Genetics

On behalf of:
